# Synthesis of Polyethylene Terephthalate (PET) with High Crystallization and Mechanical Properties via Functionalized Graphene Oxide as Nucleation Agent

**DOI:** 10.3390/molecules29091953

**Published:** 2024-04-24

**Authors:** Yingdi Dan, Yao Wang, Miaorong Zhang, Linjun Huang, Quankai Sun, Pengwei Zhang, Zengkun Li, Wei Wang, Jiangguo Tang

**Affiliations:** Institute of Hybrid Materials, National Center of International Research for Hybrid Materials Technology, National Base of International Science & Technology Cooperation, College of Materials Science and Engineering, Qingdao University, Qingdao 266071, China; danyingdi1998@163.com (Y.D.); zhangmrzhang2018@qdu.edu.cn (M.Z.); newboy66@126.com (L.H.); sunquankai11@163.com (Q.S.); pengweiz0820@163.com (P.Z.); lzk5915@163.com (Z.L.); wangwei040901@163.com (W.W.)

**Keywords:** poly(ethylene terephthalate), mechanical properties, graphene oxide, crystalline properties

## Abstract

In this work, a novel functionalized graphene oxide nucleating agent (GITP) was successfully synthesized using a silane coupling agent (IPTES), and polymer block (ITP) to efficiently improve the crystallization and mechanical performance of PET. To comprehensively investigate the effect of functionalized GO on PET properties, PET/GITP nanocomposites were prepared by introducing GITP into the PET matrix using the melt blending method. The results indicate that PET/GITP exhibits better thermal stability and crystallization properties compared with pure PET, increasing the melting temperature from 244.1 °C to 257.1 °C as well as reducing its crystallization half-time from 595 s to 201 s. Moreover, the crystallization temperature of PET/GITP nanocomposites was increased from 185.1 °C to 207.5 °C and the tensile strength was increased from 50.69 MPa to 66.8 MPa. This study provides an effective strategy for functionalized GO as a nucleating agent with which to improve the crystalline and mechanical properties of PET polyester.

## 1. Introduction

Because of its excellent thermal stability, durability, transparency, and sanitary properties, polyethylene terephthalate (PET) is widely used in various industries as a semi-crystalline thermoplastic polymer [1,2,3,4]. However, the crystallization and mechanical properties of PET limit its range of applications [5]. In recent years, research has focused on the use of organic and inorganic nucleating agents, including organic micromolecules, organic salt, montmorillonite, and carbon materials, in order to improve the crystallization of PET [6,7,8,9]. Compared with organic nucleating agents, these inorganic nucleating agents facilitate the introduction of additional crystalline nuclei, effectively shortening the nucleation cycle and improving the crystallinity of polyesters [10]. However, most inorganic nucleating agents are prone to agglomerate in PET matrix, leading to a poor dispersion of inorganic nucleating agents in PET polyester that eventually affects the properties of the composites [11].

Recently, carbon materials have attracted extensive attention as inorganic nucleating agents, such as carbon nanotubes (CNTs), carbon nanofiber, and graphene oxide (GO), that are added to PET matrices to improve the crystalline properties of PET [12,13,14]. GO is regarded as the more appropriate material for enhancing polymers due to its large specific surface area, high aspect ratio, and the way that its surface contains rich oxygen-containing groups [15,16,17]. Gao et al. have investigated the crystallization properties of PET/GO composites, and their results indicate that GO enhances the crystallization properties of PET [18]. However, in this instance the easy agglomeration of GO layers limited its application. Li et al. prepared modified GO with flexible long hydrocarbon chains as a nucleating agent with which to improve the crystallization properties of polylactide [19]. Long hydrocarbon chains improved the compatibility of GO with polylactic acid (PLA), but we found that the reaction conditions for grafting polymer chains on unmodified GO were harsh, the grafting efficiency was low, and that most of the polymer chains existed in a free form on the surface of GO and were not chemically grafted.

In this study, in order to address the previously mentioned issues, we synthesized a new functionalized GO as a nucleating agent so as to enhance the crystallization and mechanical properties of PET. Firstly, the silane coupling agent IPTES was used to modify GO, preventing agglomeration between the GO layers and improving the grafting efficiency of the block. Then, the GITP was prepared by grafting the polymer block ITP onto GO using IPTES as a linker. The polar parameters of LMPET in GITP were close to those of PET matrix, and the ITP in GITP further prevented the agglomeration of GO layers, which was conducive to the improvement of the interfacial compatibility between GITP and PET matrix. The crystallization, thermal stability, and mechanical properties of PET/GITP nanocomposites prepared via melt blending were systematically researched in order to provide a new study solution for the development of high-performance composites using functionalized GO.

## 2. Results and Discussion

### 2.1. Characterization of GITP Nanocomposites

FTIR spectra were used to verify the synthesis of GITP, as shown in Figure 1a–c. First, the synthesis of GI was confirmed via Figure 1a. Compared with the spectrum of GO, the spectrum of GI showed peaks at 2940, 1531, and 903 cm^−1^, corresponding with -CH_3_ stretching, N-H bending and Si-O-C stretching vibrations, respectively [20,21]. The -CH_3_ stretching and Si-O-C stretching vibrations derived from IPTES, and the N-H bending was formed via the reaction between the silyl alcohol group (2264 cm^−1^) of IPTES and the carboxyl and hydroxyl (3320 cm^−1^) groups of GO. The synthesis of ITP was verified via Figure 1b. Compared with the spectrum of LMPET, the broadening of -CH_2_ (2980~2870 cm^−1^) in the spectrum of TEG-LMPET was ascribed to the combined stretching vibrations of C-H in the benzene ring and CH_2_ in TEG [22]. In addition, two new absorption peaks appeared at 1531 cm^−1^ and 950 cm^−1^, corresponding with the respective N-H bending and Si-O-C stretching vibrations, in turn indicating the successful grafting of IPTES onto TEG-LMPET. As shown in Figure 1c, the spectrum of GITP contained the characteristic peaks of both GI and ITP. Compared with the spectrum of GI, the broadening of the CH_2_ (2980~2870 cm^−1^) peak in the spectrum of GITP was attributed to the grafting of ITP, which introduced a significant amount of CH_2_ groups [23,24]. Additionally, the peaks near 1129–1020 cm^−1^ were ascribed to the vibrational characteristics of Si-O-Si, and Si-O-C. The presence of a Si-O-Si peak demonstrated the condensation of GI with ITP.

The XRD patterns of GO, GI, and GITP are shown in Figure 1d. The interlayer spacing of GO, calculated based on the peak at 10.26°, was 1.00 nm. After modification with IPTES, the interlayer spacing of GI, calculated based on the peak at 8.38°, was 1.22 nm [25]. The increased interlayer spacing of GI was ascribed to silane molecules grafted onto the surface of the GO sheet. Compared with GI, the interlayer spacing of GITP, calculated based on the peak at 8.64°, was 1.37 nm, suggesting a weakening of the van der Waals forces between adjacent GO layers. The diffraction peaks of the GITP appeared at 16.3°, 17.4°, 22.36°, 25.9°, and 31.9°, corresponding with the (011), (010), (110), (100) and (101) planes of LMPET, respectively [26,27]. The results indicate that ITP was successfully grafted onto the surface of GI.

Figure 1e shows the Raman spectra of GO, GI, and GITP. Combined with the Raman fits of GI and GITP in Appendix A, the D band at around 1350 cm^−1^ represented the disordered structure (sp^3^ carbon) and the surface defects, while the G band at around 1600 cm^−1^ corresponded with the C-C stretching vibrations of the sp^2^-bonded carbon atoms [28]. The I_D_/I_G_ value (D/G intensity ratio) was employed to assess the degree of defects and chaos of carbon-based materials [29,30]. The I_D_/I_G_ value of GO was 0.89, after IPTES modification, the I_D_/I_G_ value of GI increased to 0.94, illustrating that GI enhanced the degrees of disorderliness. This was attributed to the reaction of the coupling agent with the oxygen-containing group of GO [31]. Compared with GI, the I_D_/I_G_ value of GITP slightly increased from 0.94 to 1.04, suggesting that the ITP was successfully grafted onto the surface of GI, thereby increasing the degree of disorder.

XPS was employed to further analyze the elemental composition and chemical bonding of GO, GI, and GITP, as shown in Figure 2a. The XPS spectra of the three compounds distinctly display the characteristic peaks of C1s (285.09 eV) and O1s (531.08 eV). In the spectra of GI and GITP, two new peaks emerge, corresponding to the respective Si2p (102.35 eV) and N1s (400.35 eV) peaks. The high-resolution scanning spectra of N1s and C1s were further examined for GI and GITP. The N1s spectra of GI (Figure 2b) and GITP (Figure 2e) exhibited three peaks at 401.13 eV, 399.88 eV, and 399.21 eV, corresponding with N-C=O, N-C and N-H bonds, respectively [32]. The existence of N-C=O and N-H bonds was assigned to the reaction between the isocyanate groups in IPTES and the carboxyl and hydroxyl groups present on the edges of GO. The C1s spectra of GI (Figure 2c) and GITP (Figure 3d) displayed characteristic peaks that correspond with Si-O-C (285.69 eV), C-C (284.82 eV), C=O (288.88 eV), and O-C=O (286.65 eV) bonds [33]. Relative to GI, GITP exhibited a distinct peak corresponding with the benzene (284.25eV) moiety present in LMPET. The Si2p spectrum of GITP (Figure 2f) was fitted into Si-O-Si (102.6 eV) and Si-C (102.1 eV) components, which was attributed to the condensation of silanol groups and the moiety of IPTES, respectively [34]. The above results confirm the successful progression of the grafting process.

The SEM images of GO, GI, and GITP are shown in Figure 3a–c. For pure GO (Figure 3a), the GO sheets exhibited a high level of wrinkling and folding. Relative to GO, the GI displayed a distinct layered structure, which was attributed to the fact that IPTES prevented the aggregation of GO layer sheets (Figure 3b). In Figure 3c, a large gap between the GITP sheets was observed, which was attributed to the fact that the ITP on the surface of GITP sterically prevented the condensation reaction of IPTES between different GI sheets during the drying process. Therefore, the above characteristics of GITP were favorable to the increased dispersion of GITP nanocomposites in PET matrices.

The microstructure of the synthesized GO, GI, and GITP are further analyzed by TEM, as shown in Figure 3d–f. The number of layers can be identified by the edges of the sheets. Figure 3d shows the multilayer structure of GO was agglomerated together with a relatively flat surface. As shown in Figure 3e, GI was composed of a single layer with a rough surface, indicating that the GI was not easy to agglomerate between layers [35]. The TEM image of GITP in Figure 3f shows the GITP is also basically a single layer. The TEM image of GITP confirmed the coverage of ITP on the GI surface.

### 2.2. Crystallization Properties and Thermal Stability of PET Matrix Composites

Figure 4a,b show the DSC cooling and heating curves of PET/GI with different contents of GI. The corresponding data are listed in Table 1. The glass transition temperature (T_g_) is the temperature at which the glassy state is transformed into a highly elastic state and directly affects the serviceability and processability of the material. The crystallization peak temperature (Tc), which is directly associated with the processes of nucleation and crystal growth rate, is higher, as is the crystallization rate [36]. As shown in Figure 4a, the Tc of pure PET was located at 185.1 °C. The value of enthalpy of melting (ΔH_m_) was 23.4 J/g and the value of enthalpy of crystallization (ΔH_c_) was 21.04 J/g. After adding GI, the crystallization peak was significantly enhanced, indicating that the crystallization performance of the composite PET/GI was enhanced. The Tc decreased as GI content increased to 0.6 wt%, which was attributed to the agglomeration of GO, detrimental to the crystallization of PET. The melting temperature (Tm) is related to the crystal perfection [37]. As shown in Figure 4b, The PET/GI composites exhibited a higher Tm when compared with pure PET. Notably, the maximum value of Tm was achieved when GI content was at 0.4 wt%, suggesting the crystal perfection of PET/GI composites was improved. However, the poor compatibility of GI affected the improvement of the performance of PET. To this end, GITP was used as a nucleating agent for further research. Figure 4c,d show the DSC cooling and heating curves of PET/GITP with different GITP additions, exhibiting thermal behavior similar to that of PET/GI. In addition, The Tm and Tc of the PET/GITP composites were higher than those of PET/GI. When the additive content of GITP was 0.4 wt%, both Tc and Tm reached their maximum values. The above results indicate that GITP was better than GI at improving the crystallization properties of PET. which was attributed to the grafting of the block polymer ITP, which improved the dispersion of additives in the PET matrix and prevented the agglomeration of GI layers, thereby enhancing the crystallization of composite PET materials.

The crystallization behavior of polymers can be characterized through crystallization kinetics. The relative crystallinity (X_t_) is calculated based on the theoretical melting enthalpy of 100% crystalline PET. Avrami index (n) can be derived from these figures. The shorter the half-crystallization time, the faster the rate of crystallization [38]. To provide further insights into the impact of additives on crystallization properties, the Jeziorny method was employed to analyze non-isothermal crystallization kinetics using the Avrami equation [39,40]. Figure 4e–h displays the relative crystallinity (X_t_) and crystallization time (t − t_0_), as well as the plots of ln[−ln(1 − x_t_)] against ln(t) for the nanocomposites with added GI and GITP. At an additive content of 0.4 wt%, PET/GI and PET/GITP exhibited the shortest half-crystallization time and the fastest crystallization rate. Compared with PET/GI, the PET/GITP demonstrated a higher crystallization rate. The reason for this is that the grafting of the block polymer ITP increased the dispersion of additives in the PET matrix.

The effect of additives on the crystalline properties of the PET matrix was further characterized by XRD, as shown in Figure 5a,b. The XRD patterns indicate the diffraction peaks of pure PET polyester at 16.54°, 17.63°, 22.73°, and 25.9°, corresponding with the crystal face indexes of (011), (010), (110), and (100), respectively [41]. The addition of additives did not affect the position of the PET diffraction peaks, indicating that the PET crystals remained unchanged. However, the diffraction peaks of the PET/GI and PET/GITP composites showed significant enhancement. The diffraction peaks exhibited an initial strengthening and subsequent weakening trend as the additive content increased. The highest crystallinity was observed when the addition amount was 0.4 wt%. However, the diffraction peaks of PET/GITP were sharper than those of PET/GI under the same conditions, proving that PET/GITP had better crystallization properties.

In summary, The DSC and XRD results proved to be a significant enhancement in the crystallization properties of PET/GI and PET/GITP compared with pure PET. The crystallization effect of PET/GITP was better than PET/GI. The results could be attributed to the enhanced compatibility between GITP and PET and the grafting of the block polymer ITP prevented the agglomeration of GI layers, promoting the improvement of crystallization properties in PET/GITP.

The thermal stability of thermoplastic materials directly affects their range of applications [42]. The TG curves of the prepared samples are displayed in Figure 5c. Compared with pure PET, both PET/GI and PET/GITP exhibited higher initial degradation temperatures, and the PET/GITP nanocomposite showed the highest degradation temperature. Figure 6c indicates no significant mass loss in both PET/GI and PET/GITP composites at 390 °C (<0.5%), indicating their substantial thermal stability [43]. The improvement of thermal stability was attributed to the accumulation of GI or GITP on the surface of volatile substances, inhibiting the escape of volatile substances during thermal decomposition.

### 2.3. Mechanical Properties of PET Matrix Composites

The widespread application of polyester can be attributed to its superior mechanical properties. The mechanical properties of composite materials were evaluated by tensile strength and modulus. The relevant stress–strain curves of PET nanocomposites are shown in Figure 6a,b. All prepared samples showed two stages of elastic deformation and strain hardening [44]. Figure 6c shows the tensile strength of pure PET and nanocomposites with different additions. Tensile strength reached the optimum value when the addition amount was 0.4 wt%. Because of the higher crystallinity of the PET nanocomposites doped with additives, leading to the aggregation of molecular chains to form ordered crystalline zones, the intermolecular forces become stronger and the tensile strength is also increased. The highest tensile strengths of PET/GI and PET/GITP were 65.13 MPa and 66.8 MPa, respectively. Figure 6d shows the Young’s modulus of PET, PET/GI nanocomposites, and PET/GITP nanocomposites. The addition of filler also enhanced the elastic modulus of the nanocomposites compared with pure PET. The modulus of 0.4 wt% PET/GI and 0.4 wt% PET/GITP nanocomposites reached 2.08 GPa and 2.33 GPa, respectively. Additions of fillers above 0.4 wt% led to a decrease in strength and modulus due to the interaction between the filler and the matrix being weakened.

The key to enhance the mechanical properties of PET lies in the compatibility and interface interaction between the additive and the matrix [45]. PET/GITP exhibited superior mechanical properties compared with PET/GI, primarily because of the ITP polymer block of GITP. The ITP prevented GO layer aggregation and enhanced the compatibility between GITP and PET materials, resulting in a denser internal structure. This internal structure is beneficial for the decrease of stress concentration.

## 3. Experimental Section

### 3.1. Materials

Single-layer GO was purchased from Nanjing Xianfeng Nano Material Technology Co., Ltd. (Nanjing, China). Ethylene glycol (EG), antimony trioxide (Sb_2_O_3_), phenol, and tetrachloroethane were purchased from Sinopharm Chemical Reagent Co., Ltd. (Shanghai, China). Triethylene glycol (TEG), *N*,*N*-Dimethylformyl (DMF) and dibutyltin dilaurate (DBDU) were purchased from Shanghai Macklin Biochemical Technology Co., Ltd. (Shanghai, China). Ethanol and 3-isocyanatopropyl-triethoxysilane (IPTES) were purchased from Shanghai Aladdin Biotechnology Co., Ltd. (Shanghai, China) The commercial PET polyester slices (processing grade: fiber grade, non-extinction, intrinsic viscosity 0.8 dL/g) were purchased from Sinopec Yizheng Chemical Fiber Co., Ltd. (Yizheng, China).

### 3.2. Synthesis of GI Nanocomposites

The synthesis process of GI nanocomposites is shown in Figure 7a. An amount of 0.6 g of GO nanosheets were ultrasonically dispersed in 300 mL of DMF solvent to form a uniform GO suspension at room temperature. Then, 10 mL of IPTES and 0.5 mL of DBDU were added to the GO suspension. The reaction was performed while stirring at 100 °C for 6 h. Thereafter, the resulting mixture was washed four times by centrifugation with absolute ethanol. Finally, GI powder was obtained after drying the product in a vacuum oven.

### 3.3. Synthesis of Low Molecular Block Polymers (ITP)

The synthesis process of ITP is shown in Figure 7b. LMPET was prepared using the transesterification method reported previously [46]. For the synthesis of TEG–LMPET, 18.3 g of LMPET and 6 mL of EG were dissolved in phenol/tetrachloroethane (mass ratio 1:1). The mixture was stirred for 1 h at 60 °C. Then, 32 mL of TEG and 30 mg polycondensation catalyst of Sb_2_O_3_ were added to the mixture solution under stirring at 60 °C for 1 h. After cooling, The TEG–LMPET was obtained via rinsing with phenol/tetrachloroethane (mass ratio 1:1) and absolute ethanol until the excessive reactants were removed and dried at 60 °C in a vacuum oven for 24 h.

For the synthesis of ITP, 1 g of TEG-LMPET was dissolved in phenol/tetrachloroethane (mass ratio 1:1). The mixture was stirred for 1 h at 100 °C. Then, 5 mL of IPTES and 0.5 mL of DBDU were added into the mixture solution under stirring at 100 °C for 10 h. After cooling, the resulting mixture was washed four times by centrifugation with absolute ethanol. Finally, the product ITP was dried at 60 °C for 24 h.

### 3.4. Synthesis of GITP Nanocomposites

The synthesis process of ITP is shown in Figure 7c. Amounts of 20 mg of GI and 50 mg of ITP were ultrasonically dispersed in 30 mL of anhydrous ethanol to form a suspension at room temperature. Then, 30 mL of deionized water was added to the aforementioned suspension, the mixtures were stirred at 60 °C for 12 h. Finally, the mixture was washed three times with water/ethanol solution (1:1 *v*/*v*). Finally, the product GITP was dried at 60 °C for 24 h.

### 3.5. Preparation of PET/GI and PET/GITP Nanocomposites

The GI, GITP, and PET polyester chips were dried at 100 °C for 48 h in a vacuum oven. Then, PET/GI and PET/GITP nanocomposites were obtained by melt-blending PET polyester chips with GI and GITP using a micromixer. Finally, PET polyester chips, PET/GI, and PET/GITP were introduced into the twin-screw micro-composite extruder and injection molding machines, respectively. PET/GI and PET/GITP, with different contents of GI and GITP (0.2, 0.4, and 0.6 wt%), were prepared. The effects of different additive contents on the properties of PET materials were then compared.

### 3.6. Characterization

The surface functional groups and the chemical composition of the samples were characterized by Fourier transform infrared (FT-IR, IS-50, Thermo-Fisher, Waltham, MA, USA) in the range of 4000–400 cm^−1^ and X-ray photoelectron spectroscopy (XPS, K-Alpha, Thermo-Fisher, USA). Raman spectra of the samples were obtained on a Horiba LabRam HR800 Raman spectrometer with 532 nm laser excitation. The crystallinity of the polymer was tested by X-ray diffraction (XRD, D8 advance, Bruker AXS, Karlsruhe, Germany). The physical morphology and structure of GO, GI, and GITP were characterized by scanning electron microscopy (SEM, JSM-7500F, JEOL, Tokyo, Japan) and transmission electron microscopy (TEM, JSM-2100 F, JEOL, Tokyo, Japan).

The crystallization and melting performance of the samples were investigated by a differential scanning calorimeter (DSC, DSC250, Waters, Milford, MA, USA) in a N_2_ atmosphere. PET, PET/GI, and PET/GITP composites were heated at a rate of 30 °C min^−1^ from room temperature to 300 °C and maintained for 10 min to eliminate thermal history. Then, they were cooled to 30 °C at 10 °C min^−1^ to obtain the crystallization curve before being heated to 300 °C at 10 °C min^−1^ to obtain the melting curve. The thermal properties of the samples were investigated by a thermal analyzer (TA, SDT650, Waters, USA) in a N_2_ atmosphere. The samples were heated from room temperature to 600 °C at 10 °C min^−1^. Tensile tests of the samples were obtained by a universal material testing machine (Instron 3382, Massachusetts, USA).

## 4. Conclusions

Here, a novel nucleating agent GITP was designed, and its effects on the crystallization and mechanical properties of PET were explored systematically. Owing to the admirable dispersivity of GITP within the PET matrix, GITP had a good effect on the improvement of the crystallization properties, thermal stability, and mechanical properties of PET/GITP nanocomposites. The GITP with a 0.4 wt% addition was able to significantly enhance the crystallization properties and thermal stability of PET, increasing the crystallization temperature and the melt temperature. In particular, the relative crystallinity increased by 105.6%, and the half-crystallization time decreased from 595 s to 201 s. At the same concentration level, the tensile strength of PET/GITP was increased from 50.7 MPa to 66.8 MPa and the tensile modulus of 2.33 GPa was increased by 60.7%. The present work provides a new idea for the development of functionalized GO in order to improve the crystallization and mechanical properties of PET and represents a new strategy for the development of more novel functional polyester composites.

## Figures and Tables

**Figure 1 molecules-29-01953-f001:**
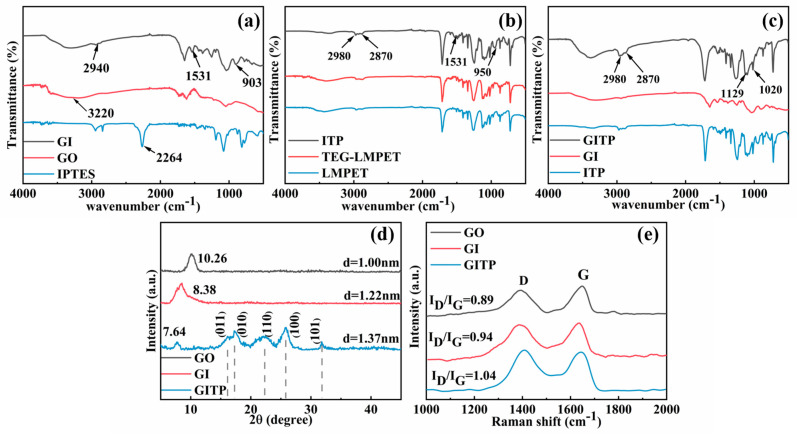
FTIR spectra of (**a**) GI, GO, and IPTES; (**b**) ITP, TEG-LMPET, and LMPET; (**c**) GITP, GI, and ITP; (**d**) XRD patterns of GO, GI, and GITP; and (**e**) Raman spectra of GO, GI, and GITP.

**Figure 2 molecules-29-01953-f002:**
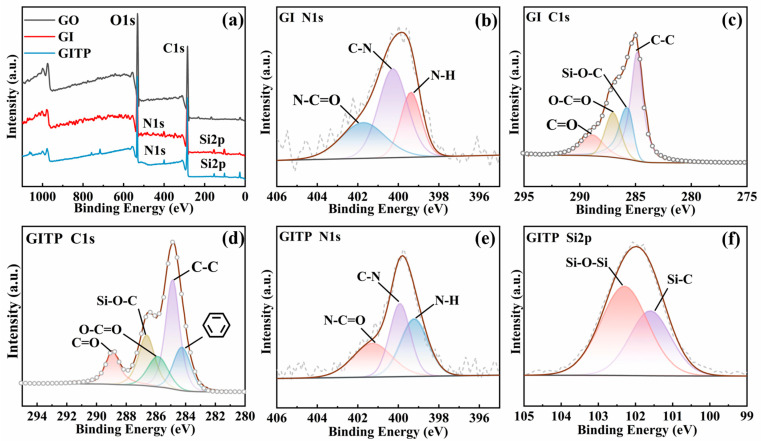
(**a**) XPS spectra of GO, GI, and GITP; (**b**) N1s and (**c**) C1s spectra of GI; (**d**) C1s, (**e**) N1s and (**f**) Si2p spectra of GITP.

**Figure 3 molecules-29-01953-f003:**
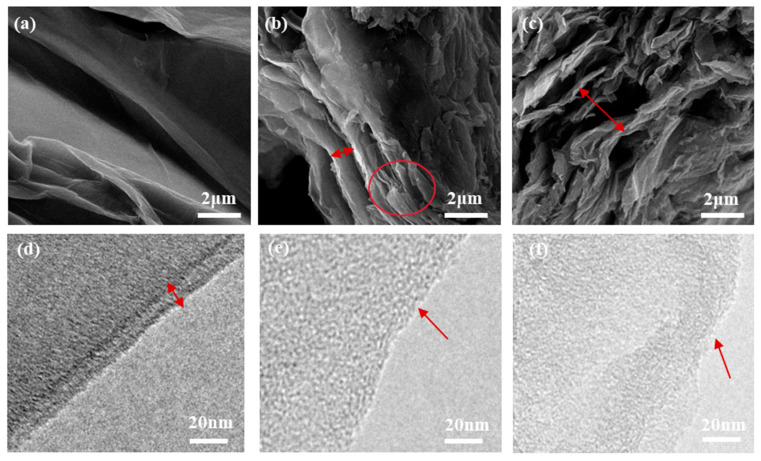
SEM images of (**a**) GO, (**b**) GI, and (**c**) GITP. TEM images of (**d**) GO, (**e**) GI, and (**f**) GITP.

**Figure 4 molecules-29-01953-f004:**
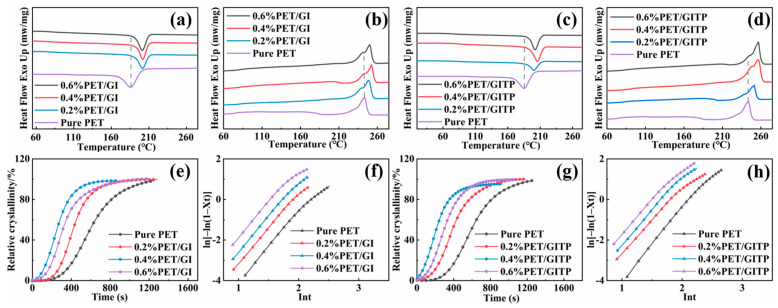
Differential scanning calorimeter cooling (**a**,**c**) and heating (**b**,**d**) curves for (**a**,**b**) PET/GI and (**c**,**d**) PET/GITP with different additive concentrations. The relative crystallinity (X_t_) versus crystallization time (t − t_0_) of (**e**) PET/GI and (**g**) PET/GITP and the logarithm Avrami diagram ln[−ln(1 − X_t_)] versus ln(t) of (**f**) PET/GI and (**h**) PET/GITP.

**Figure 5 molecules-29-01953-f005:**
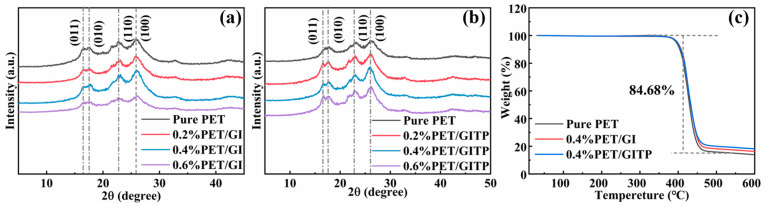
The XRD spectra of (**a**) PET/GI and (**b**) PET/GITP with varying contents. (**c**) Thermal analysis of hybrid PET materials.

**Figure 6 molecules-29-01953-f006:**
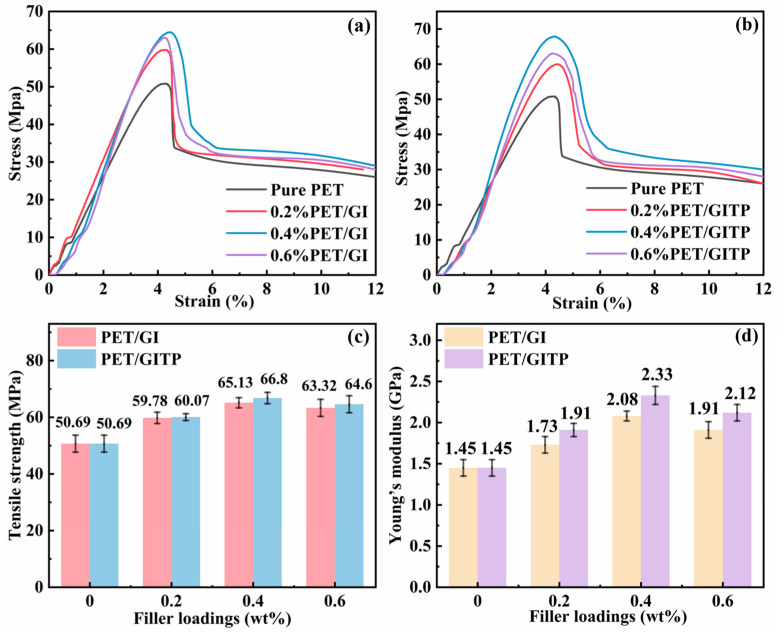
Stress–strain curves of (**a**) PET/GI and (**b**) PET/GITP with different additive contents. (**c**) Data graph of tensile strength and (**d**) modulus of PET/GI and PET/GITP.

**Figure 7 molecules-29-01953-f007:**
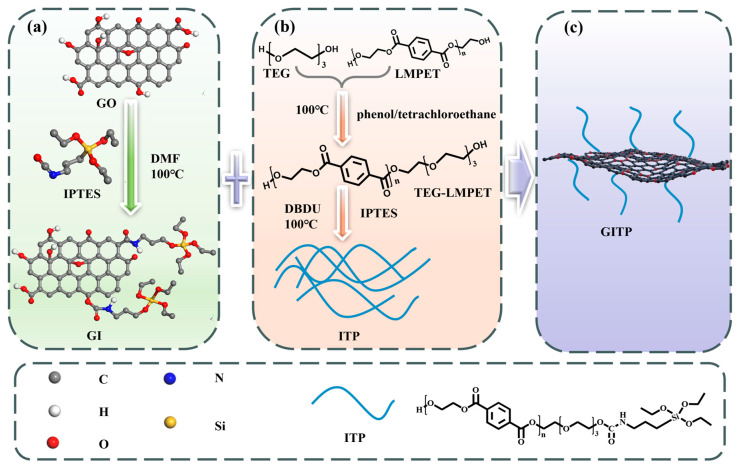
Synthesis of (**a**) GI nanocomposite, (**b**) ITP, and (**c**) GITP nanocomposite.

**Table 1 molecules-29-01953-t001:** Differential scanning calorimetry data of GI-doped and GITP-doped complex additive-modified polyethylene terephthalate (PET) hybrid materials.

Content (%)	T_c_ (°C)	T_m_ (°C)	T_g_ (°C)	ΔH_m_ (J/g)	ΔH_c_ (J/g)	X_t_ (%)	T_1/2_ (s)	n
0	185.1	244.1	73.63	23.4	21.04	16.1	595	2.7
0.2%PET/GI	196.9	249.1	74.9	36.65	28.36	26.2	415	2.63
0.2%PET/GITP	201.13	251	75.2	33.89	26.24	24.2	388	2.57
0.4%PET/GI	202.3	252.5	75.82	42.87	40.64	30.6	250	2.52
0.4PET/GITP	207.5	257.1	78.32	46.35	42.01	33.1	201	2.45
0.6%PET/GI	199.1	250.2	73.97	38.7	30.89	27.7	314	2.66
0.6%PET/GITP	203.6	355.8	74.01	42.09	33.99	30.1	299	2.60

## Data Availability

Data are contained within the article.

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
