# Peer review of "Synthesis of Polyethylene Terephthalate (PET) with High Crystallization and Mechanical Properties via Functionalized Graphene Oxide as Nucleation Agent"

_molecules, 2024, doi:10.3390/molecules29091953_

Round 1
Reviewer 1 Report
Comments and Suggestions for Authors
The PET polymer has very wide potential for application. This manuscript contributes to new method of crystallization. The manuscript deserves to be published after minor revisions.
I suggest to fit the DSC and Raman and add it to supplementary data (Just normal line to the X asis is not sufficient for definition of values).
This sentence could be heated up by more references “The ID/IG value (D/G intensity ratio) was employed to assess the degree of defects and chaos of carbon-based materials [30; DOI: 10.3390/polym12122766 ].“
Could you please provide more detailed explanation of mechanical properties?
Comments on the Quality of English LanguageEnglish is fine.
Author Response
Dear Editors and Reviewers:
Thank you for your decision and constructive comments on my manuscript. We have carefully considered the suggestion of Reviewer and make some changes. The parts of the manuscript in red are revisions based on your comments. The instructions for point-to-point revisions are as follows:
Comments 1: I suggest to fit the DSC and Raman and add it to supplementary data (Just normal line to the X asis is not sufficient for definition of values).
Response 1: Thank you for pointing this out. I agree with this comment. Therefore, I have fitted the Raman data for GI and GITP and added it to the support material. At the same time I added a description of the DSC data. You can see the additional figures at the end of the manuscript.
Comments 2: This sentence could be heated up by more references “The ID/IG value (D/G intensity ratio) was employed to assess the degree of defects and chaos of carbon-based materials [30; DOI: 10.3390/polym12122766 ].“
Response 2: We sincerely appreciate your valuable comments. We have carefully checked the literature and added more references on ID/IG . You can see the modification here in lines 160-161
Comments 3: Could you please provide more detailed explanation of mechanical properties?
Response 3: Thank you for pointing this out. I agree with this comment. Therefore, I have added to the formulation of mechanical properties.You can see the modification on line 285-288.
Again, thank you for allowing us to strengthen our manuscript with your valuable comments and queries. We have worked hard to incorporate your feedback and hope that these revisions persuade you to accept our submission.
The revised manuscript is attached as a Word file.
Yours sincerely
Yao Wang
13-Apr,2024
Qingdao University
Reviewer 2 Report
Comments and Suggestions for Authors
The manuscript is devoted to the study of Polyethylene Terephthalate (PET), the most common representative of the polyester class. PET is widely used in various fields of human activity: the production of threads and fibers, containers of various types, sheet material that is transparent to sunlight (including UV) and resistant to environmental influences.
Therefore, the research aimed at improving the thermal stability, crystalline and mechanical properties of PET polyester when using functionalized GO (graphene oxide) as a functionalized graphene oxide nucleating agent (GITP) is certainly relevant.
The authors developed a new nucleating agent GITP and systematically investigated its effect on the properties of PET using complementary modern methods and equipment: Fourier transform infrared (FT-IR), differential scanning calorimeter (DSC), thermal analyzer (TA), X-ray photoelectron spectroscopy (XPS), Raman spectrometry, X-ray diffraction (XRD), scanning electron and transmission electron microscopy (SEM and TEM).
The most interesting scientific results of the study are a significant improvement in the thermal stability, crystallization properties and mechanical properties of PET/GITP compared to pure PET.
The weakness of the paper can be attributed to some inaccuracies and misprints in the presentation, which make it difficult to understand the results obtained.
Below are specific comments on the article
Line 43
the abbreviation PLA should be disclosed in the text
Line 188-189
“For pure GO (Fig. 4a), The GO sheets…..”
Should be
“…the GO sheets…”
Lines 187-195
“a distinct layered structure” is not obvious in Fig 4b. More explanations are required. The layered structure should be indicated in the image.
“.. a large gap between the GITP sheets…” is not obvious in Fig 4c. More explanations are required. The large gap between the GITP sheets should be indicated in the image. Fig. 4b and 4c look quite similar. More explanations are required.
Lines 196-202
The TEM images discussion requires more explanation to prove the statements in the text.
In particular,
Line 197
“Fig.4d shows the multilayer structure of GO was agglomerated together with a relatively flat surface”
There are two areas in the image. Where is the agglomerated multilayer structure of GO?
Line 198
“As shown in Fig. 4e, GI was composed of a single layer with a rough surface,…
Thera are two areas in the image. How do you define the single layer in the image?
Line 200
“Fig. 4f shows the GITP exhibited a relatively transparent and thin morphology with a smaller size”
Thera are two areas in the image. Which area belongs to GITP with thin morphology? What is “thin morphology”?
In general, Figure 4 is redundant and contains unclear information. I suggest removing Figure 4 from the text.
Line 241
All parameters specified in Table 1 must be disclosed and explained in the text.
What do the parameters Tg, ΔHm, ΔHc and n mean in Table 1?
Lines 251-252
“However, the diffraction peaks of the PET/GI and PET/GITP composites showed significant enhancement. The sharper the diffraction peak, the higher the crystallinity of the sample”
This statement contradicts a similar parameter in Table 1 (Xt), which is up to 33% for 0.4PET/GITP. Additional explanation is required.
The sharpness of the diffraction peak is a full width at half maximum (FWHM). You need to provide FWHM values, at least for sharpest diffraction peaks of the PET/GI and PET/GITP.
Line 300
“ ..the relative crystallinity reached 105.6%,…”
This statement contradicts with the same parameter in the Table 1 (Xt) which is up to 33% for 0.4PET/GITP. More explanation is required.
Author Response
Dear Editors and Reviewers:
Thank you for your decision and constructive comments on my manuscript. We have carefully considered the suggestion of Reviewer and make some changes. The parts of the manuscript in red are revisions based on your comments. The instructions for point-to-point revisions are as follows:
Comments 1: the abbreviation PLA should be disclosed in the text
Response 1: Thank you for pointing this out. I agree with this comment. Therefore, I added an explanation of PLA.You can see the modification on line 43.
Comments 2: “For pure GO (Fig. 4a), The GO sheets…..” Should be “…the GO sheets…”
Response 2: Thank you for pointing this out. I agree with this commen. You can see the modification on line 190.
Comments 3: “a distinct layered structure” is not obvious in Fig 4b. More explanations are required. The layered structure should be indicated in the image. “.. a large gap between the GITP sheets…” is not obvious in Fig 4c. More explanations are required. The large gap between the GITP sheets should be indicated in the image. Fig. 4b and 4c look quite similar. More explanations are required.
Response 3: Thank you for pointing this out. I agree with this comment. Therefore, I have redone the test, replacing the SEM image with one where the GI and GITP lamellar structures are more pronounced, and have marked the image.You can see the modifications in Figure 4b-c.
Comments 4: “Fig.4d shows the multilayer structure of GO was agglomerated together with a relatively flat surface”
Response 4: Thank you for pointing this out. I agree with this comment. Therefore, I have explained how the layer structures were observed, and I have also modified and marked them in Figure 4d.You can see the modifications in Figure 4d and line 198-199 .
Comments 5: “As shown in Fig. 4e, GI was composed of a single layer with a rough surface,…
Thera are two areas in the image. How do you define the single layer in the image”
Response 5: Thank you for pointing this out. I agree with this comment. Therefore, I have marked the edges of the GI in Figure 4e. You can see the modifications in Figure 4e.
Comments 6: “Fig. 4f shows the GITP exhibited a relatively transparent and thin morphology with a smaller size” Thera are two areas in the image. Which area belongs to GITP with thin morphology? What is “thin morphology”?
Response 6: Thank you for pointing this out. I agree with this comment. Therefore, I have modified the morphological description of GITP by removing the controversial phrase "Fig. 4f shows the GITP exhibited a relatively transparent and thin morphology with a smaller size". It was also marked in Figure 4f, and you could see the modifications in Figure 4f and in line 201-202.
Comments 7: “All parameters specified in Table 1 must be disclosed and explained in the text.
What do the parameters Tg, ΔHm, ΔHc and n mean in Table 1”
Response 7: Thank you for pointing this out. I agree with this comment. Therefore, I have explained the meaning of Tg, ΔHm, ΔHc and n, which you can see on line 210-212, 215-216, 236.
Comments 8: “The sharpness of the diffraction peak is a full width at half maximum (FWHM). You need to provide FWHM values, at least for sharpest diffraction peaks of the PET/GI and PET/GITP.”
Response 8: Thank you for pointing this out. I agree with this comment. Therefore,I modified the description of the XRD of the nanocomposites. You can see the modification on line 256-259.
Comments 9: “ ..the relative crystallinity reached 105.6%,…” This statement contradicts with the same parameter in the Table 1 (Xt) which is up to 33% for 0.4PET/GITP. More explanation is required.
Response 9: Thank you for pointing this out. I agree with this comment. Therefore I have modified the wrong expression. The correct expression is: the relative crystallinity has increased by 105.6%. You can see the modification on line 305-306.
Again, thank you for giving us the opportunity to strengthen our manuscript with your valuable comments and queries. We have worked hard to incorporate your feedback and hope that these revisions persuade you to accept our submission.
The revised manuscript is attached to the Word file.
Yours sincerely
Yao Wang
13-Apr,2024
Qingdao University